1    **Analysis of improvements in MOPITT observational coverage over Canada**

5    Heba S. Marey[1], James R. Drummond[1], Dylan B. A. Jones[1], Helen Worden[2], Merritt N. Deeter[2],
6                                   John Gille[2], and Debbie Mao[2]

8    1 University of Toronto, Department of Physics, Atmospheric Science Group, Ontario, Canada, 2 National Center for Atmospheric Research,

9                                              Boulder, Colorado, USA.

11                                                   Abstract

13        The Measurements of Pollution in the Troposphere (MOPITT) satellite instrument has been
measuring global tropospheric carbon monoxide (CO) since March 2000, providing the longest
nearly continuous record of CO from space. During its long mission, the data processing algo-
rithms have been updated to improve the quality of CO retrievals and the sensitivity to the lower
troposphere. Currently, MOPITT retrievals are only performed for clear-sky observations or over
low clouds for ocean scenes. The cloud detection scheme was modified in the new V9 product,
resulting in an improvement in observational coverage, especially over land. Comparison of the
spatial and seasonal variations of the data coverage in V9 and V8 shows differences with
significant geographical and temporal variability, with some regions such as Canada and the Am-
azon exhibiting a doubling of data in winter. Here we conducted an analysis of Moderate Resolu-
tion Imaging Spectroradiometer (MODIS) cloud heights and cloud mask products along with
MOPITT retrieval cloud flag descriptors to understand the impact of cloud conditions on the
MOPITT observational coverage, with a particular focus on observations over Canada. The
MOPITT CO total column (TC) data were modified by turning off the cloud detection scheme to
allow a CO retrieval result regardless of their cloud status. Analyses of the standard V8 CO TC
product (cloud filtered) and non-standard product (non-cloud masked) were conducted for selected
days. Results showed some coherent structures that were observed frequently in the non-masked
CO product that was not present in the V8 product and could potentially be actual CO features.
Many times, these CO plumes were also seen in the Infrared Atmospheric Sounding Interferometer

(IASI) CO TC product. The MODIS cloud height analysis revealed that a significant number of low cloud CO retrievals were discarded in the V8 product. Most of the missed CO plumes in the V8 product are now detected in the new V9 product as a result of the dependence of MOPITT radiance ratio (MRT) test over land. Comparisons of the MRT and MODIS cloud height data indicate a remarkable negative correlation. As a result of modified V9 cloud detection algorithm, a significant portion of the low cloud CO retrievals is now incorporated in the new V9 MOPITT product. Consequently, the observational coverage over Canada is significantly improved, which benefits analyses of regional CO variability, especially during extreme pollution events. We also conducted a comparison of MOPITT and IASI CO TC and found generally good agreement, with about a 5-10% positive bias that is more pronounced in highly polluted scenes.

## 1. Introduction

Carbon monoxide (CO) in the atmosphere has a medium lifetime (weeks to months), which is long enough to track atmospheric physical and chemical processes over a range of spatial scales from space (Jiang et al., 2011, Edwards et al., 2006; Duncan et al., 2007). Hence, satellite measurements of atmospheric CO are useful for studying both transported and local sources of pollution as well as atmospheric chemistry.

The Measurements of Pollution in the Troposphere (MOPITT) satellite instrument provides the longest dataset of CO from space. It has been measuring tropospheric CO using gas filter correlation radiometry (GFCR) since March 2000 (Drummond et al., 1996, Drummond et al., 2010, Deeter et al., 2017), with a footprint of 22 km × 22 km and global coverage every 3 days (Deeter et al., 2003). It is on board the Terra satellite, which is in a sun-synchronous polar orbit at 705 km of altitude and crosses the equator at 10:30 local time (Drummond et al., 1996). Furthermore, it is the only satellite instrument that measures CO in both the thermal infrared (TIR, 4.7 μm) and near infrared (NIR, 2.3 μm). This long-term data record provides a unique opportunity for analyzing interannual variability and long-term trends in the distribution of CO, atmospheric transport, and tropospheric chemistry that are associated with human activity and climate change (Worden et al., 2013; Strode et al., 2013, Buchholz et al., 2021).

During MOPITT's long mission, data processing algorithms have been updated considerably to improve the quality of the CO retrievals and their sensitivity to the lower troposphere. However,

MOPITT cannot "see" through cloud and this represents a significant obstruction to measurement
spatial coverage. The current cloud detection algorithm, using both MOPITT and Moderate Reso-
lution Imaging Spectroradiometer (MODIS) information (Warner, et al., 2001), rejects pixels with
a significantly amount of cloud cover, thereby reducing the number of pixels retrieved. This leads
to global maps with gaps in CO data where clouds are present.
Retrieving CO gas in cloudy conditions represents a major challenge. The presence of clouds
in the observed scene enhances reflectivity and blocks the atmosphere below the clouds for cloudy
scenes compared to cloud-free sky scenes. The albedo and in-cloud absorption effects enhance the
sensitivity to trace gases above the clouds, while the shielding effect impacts the vertical sensitivity
of the measurement which results in an inaccurate estimation of the trace gas column. Various
techniques have been proposed to cope with this problem depending on the spectral range of the
measurements. These techniques can be grouped into the following four approaches.
The first approach is the threshold method, where only observations under clear sky conditions
or weakly cloud contaminated scenes (determined by using threshold-based algorithms to detect
clouds and develop cloud masks) are considered (Ackerman et al., 1998; Deeter, 2003; Warner, et
al., 2001). The second approach, referred to as cloud clearing, is to reconstruct clear column radi-
ances that would have been present if there were no clouds. Cloud clearing is used for Atmospheric
Infrared Sounder (AIRS) atmospheric CO retrievals where a reconstructed pixel consisting of a 3
x 3 array (9 pixels are used) is produced, resulting in 45 km spatial resolution (Susskind et al.,
2003; Li et al., 2005). Both of these approaches avoid the need for complex modeling of cloud
effects, but have the added complexity of characterizing errors resulting from un-modeled cloud
fields. The third approach is to solve for the radiative effects of clouds directly in the inversion
process. This approach is used for retrieving profiles (Kulawik et al. 2006) from measurements
from the Tropospheric Emission Spectrometer (TES). The fourth approach is utilized for CO re-
trievals over land and ocean in the presence of low-altitude clouds from measurements from the
TROPOspheric Monitoring Instrument *(TROPOMI)*. In this approach, shortwave infrared (SWIR)
measurements of methane TC are used to filter out observations with high and optically thick
clouds to retrieve the trace gas information (Vidot et al. 2012, Landgraf et al., 2016).
For MOPITT, due to the lack of spectral information and collocated methane data, only the
first two approaches are possible and, unfortunately, the results of the reconstructed clear column
radiances using two adjacent pixels are not sufficiently precise for viable retrievals. Consequently,
adjustments to the current MOPITT cloud detection scheme is the only one of the four approaches
that can be employed.
Deeter et al., (2021) recently made significant changes to the cloud detection scheme resulting
in a new MOPITT product V9. Those changes impacted the MOPITT coverage rate, especially
over land. Hence, the aim of this study is to conduct an analysis of MODIS cloud heights and cloud
mask products along with MOPITT retrieval cloud flag descriptors to understand the impact of
cloud conditions on the MOPITT observational coverage, with a particular focus on observations
over Canada.

## 2.  Data and Methodology
This study uses data from three satellite instruments, MOPITT, IASI, and MODIS. MODIS,
and MOPITT are all onboard the Terra satellite (with an equatorial crossing time of 10:30 am local
time (LT)), which facilitates the collocation of observations in space and time. IASI on MetOp-A
has an equatorial crossing time of 9:30 am LT.
### 2.1 MOPITT
The MOPITT instrument is onboard the NASA Terra, which is a Sun-synchronous polar-
orbiting satellite. It has a spatial resolution of $22 \times 22$ km with a swath width of 640 km which
covers the globe every 3–4 days.  MOPITT Version 8 and Version 9 (V8 and V9) Level 1 (L1)
and Level 2 (L2) TIR products are used in this study. L1 data corresponds to all of the radiance
observations that are obtained in MOPITT swaths. They are used subsequently as input to the
algorithms that retrieve the CO vertical profiles and total column (TC) amounts, which are referred
to as L2 data. The MOPITT L2 products that are utilized here are the CO total column (TC) abun-
dances and two cloud diagnostics contained in the MOPITT L2 files: the MOPITT cloud descrip-
tion index and the MODIS cloud diagnostics vector.

### 2.2 MODIS
The MODIS products used in this study are the Collection-6 1-km cloud mask (MOD35)
and the cloud height 5-km resolution (MOD06) data. MODIS measures radiances at 36 wave-
lengths, including infrared and visible bands with spatial resolution from 250 m to 1 km. The
MODIS cloud mask algorithm uses up to 19 MODIS spectral bands for better cloud detection
(Ackerman et al., 2008, 1998). The MODIS cloud height is derived using 5 thermal infrared bands
(both day and night) at 5 km spatial resolution.

**2.3 IASI-A**

IASI-A is a Fourier Transform Spectrometer on the European space agency (EPS)/MetOp-A

satellite launched in 2006 with a spectral coverage range from 3.62 to 15.5 μm (645 to 2760 cm−1)
including the CO 2140 cm$^{-1}$ TIR band. It views the ground through a cross-track rotary scan mirror
with a horizontal resolution of 12 km diameter at nadir, which increases at the larger viewing
angles. The width of the swath is ~ 2200 km with a total of 120 views. The IASI instrument takes
measurements day and night which gives a global coverage twice a day with some gaps between
orbits around the equator. However, clouds in the field of view can obstruct the measurements and
hence reduce the number of the observations (Clerbaux et al., 2009). This study used L2 IASI-A
CO TC values that were retrieved by LATMOS (Laboratoire Atmosphères, Milieux, Observations
Spatiales) using a retrieval code, FORLI (Fast Optimal Retrievals on Layers for IASI), developed
at ULB (Université Libre de Bruxelles) (https://iasi.aeris-data.fr/co/).  Data are retrieved for a
cloud fraction of less than 25 % (Clerbaux et al., 2009).

**3.   MOPITT cloud detection scheme**

The MOPITT retrieval algorithm only performs retrievals in clear-sky conditions. The

MOPITT procedures for identifying clear-sky retrievals from cloud-contaminated pixels involves
a threshold method that makes use of two independent tests, (1) a MOPITT radiance ratio threshold
and (2) a MODIS cloud mask threshold within the MOPITT field of view (Warner et al., 2001,
Deeter, 2011), which are described below.

**MOPITT radiance threshold**. Radiance from the MOPITT 4.7 μm thermal channel radiance is
compared to the a priori clear-sky radiance calculated by The MOPITT Operational Fast Forward
Model (MOPFAS) (Edwards et al., 1999) for each pixel. If the measured/calculated radiance ratio
is ≥1.0 for V8 and V9 and ≥ 0.955 for other versions (V7 and before), then the observation is
considered "clear". For this test however, the threshold value may be exceeded under temperature
inversion conditions where clouds are warmer than the underlying surface. This threshold method
is not applicable to polar regions due to the frequent temperature inversions at night, and to avoid
the effect of possible snow and ice coverage on the daytime signals (Warner et al., 2001).
**The MODIS Cloud threshold**. The MODIS swath (2330 km) is much wider than the MOPITT
swath (640 km), so it provides complete overlap for MOPITT passes. The MODIS cloud mask
(MOD35 L2) product (Ackerman et al., 2008) that is used here has 1 km horizontal resolution at
nadir (Ackerman et al., 1998). Therefore, each MOPITT pixel can encompasses ~ 480 MODIS 1
x 1 km pixels. After co-location, relevant MODIS cloud mask parameters of the MODIS are gath-
ered and averaged for each MOPITT pixel. MOD35_L2, containing data collected from the Terra
platform is used to get the cloud count at each MOPITT pixel and if the MODIS cloud percent is
less than 5%, then the MOPITT pixel is considered clear.

In the previous MOPITT products (V8 and before), the MODIS test value supersedes the

MOPITT value over land, i.e., if the MODIS test is "clear" and the MOPITT test is "cloudy", then
the MOPITT pixel will be considered "clear" (Warner et al., 2001, Marey et al., 2018). However,
if the MOPITT test identifies the pixel as clear and the MODIS test identifies the pixel as cloudy,
then a low cloud test is done. The low cloud test exploits the MODIS IR and visible reflectance
(Warner et al., 2001: Deeter et al., 2017). To assign low clouds for daytime observations, an aver-
aged MODIS IR threshold test value should be ≥0.9 and an averaged MODIS visible reflectance
test value should be ≤0.95. For nighttime observations, a MODIS IR temperature difference test
value ≥0.9 is interpreted as low clouds (Warner et al., 2001, Marey et al., 2018). While for ocean
scenes even if the low cloud test did not pass, the pixel is considered clear based on either the
MOPITT or MODIS test result (Deeter et al., 2017).

For the V9 product, the modified cloud detection algorithm (Deeter et al., 2021) allows CO

retrievals over land when the MOPITT radiance ratio test indicates the pixel is clear although the
MODIS cloud mask test assigns the pixel as cloudy. Hence, a cloud index value of 6 (Table 1) is
now applied for both ocean and land areas (Deeter et al., 2017 and 2021).

The final clear/cloudy decision for each MOPITT pixel is based on set of rules summarized

in six cloud indices as follows: The pixel is assigned to be clear and hence retrieved if:
1: MODIS data are missing but the MOPITT radiance threshold is passed (rare).
2: MODIS data are clear and the MOPITT radiance threshold is passed. (most confidently clear)
3: MODIS data are clear but MOPITT radiance threshold is failed. The MODIS result overrides
the MOPITT result.
4: MODIS data are cloudy but the MOPITT radiance threshold is passed. In this case, the MODIS
low cloud test is applied and in the case of a low cloud, the pixel is treated as clear (occurs mostly
over ocean scenes).
5: Polar regions only (> 65° N or S latitude): MODIS data are clear. MOPITT test is not used.
6: Ocean and land scenes for V9 no MODIS low cloud: MODIS data are cloudy and the
MOPITT radiance threshold is passed. This was introduced in V7 for ocean scenes to correct for
an observed degradation in MODIS cloud products (Moeller and Frey, 2017).
If the pixel does not pass any of these tests, then no retrieval is performed. The six cloud
indices are reported in Level 2 MOPITT files in the "Cloud Description" diagnostic as presented
in Table 1.

Table 1. MOPITT Cloud Descriptor Values in L2 CO retrievals

| Descriptor value | MOPITT assignment | MODIS assignment | Notes |
|---|---|---|---|
| 1 | clear | missing | MODIS data are not available |
| 2 | clear | clear | |
| 3 | cloudy | clear | |
| 4 | clear | cloudy, low clouds | |
| 5 | Not used | clear | Used only in polar regions |
| 6 | clear | cloudy, no low clouds | Introduced in MOPITT V7, for ocean observations only |



**4.  Results and Discussion**
**4.1  Assessment of the successful MOPITT CO retrievals**
To assess the successful MOPITT CO retrievals in terms of data coverage, the statistics of
the L2 data from 2000 to 2020 for V9 and V8 are computed. Buchholz et al. (2017) recommended
avoiding the use of MOPITT above 60°N as the sea ice may not be correctly accounted for in the
retrievals. The fraction of daily valid data between 90°S–90°N and 60°S–60°N (for land and ocean
combined) are shown in Figure 1. The successful rate is calculated by taking the ratio of the num-
ber of daily CO data retrievals (L2) to the total number of daily radiance measurements (L1). For
the 90°S–90°N, the successful retrieval rate of V8 and V9 varies between 27%–33% and 35%–
40%, respectively. While the 60°S–60°N domain has a successful retrieval rate between 34%–
42% and 40%–50%, for V8 and V9 respectively. Therefore, the number of daytime V9 MOPITT
retrievals has increased by 15-20% for 90°S–90°N and 60°S–60°N relative to Version 8 product.
However, the gain in data coverage varies significantly on spatial level.

Figures 2-5 show the seasonal spatial coverage rate per day (the faction of the successful

retrievals, L2 to the total number of radiance measurements, L1) using 2014 as a representative
year gridded in 1° × 1° bins. The top and the bottom figures present V9 and V8 daily coverage
rate, respectively while the middle ones indicate V9-V8 percent   It is apparent that some regions
exhibit high coverage rates (close to 100%) in all seasons for both V8 and V9, such as northern
Africa, so there are no added observations over such regions as it is indicated by the middle panels
(V9-V8).  While other regions exhibit large gain in retrievals compared to V8 product which varies
seasonally. For example, in Canada, the data coverage of V8 (bottom panel of Figure 4) reached
50% in summer (e.g. Hudson Bay), but drops to less than 10% in winter (bottom panel of Figure
2) due to high cloud cover. Interestingly, V9 successful retrievals (top panels) for Canada demon-
strated significant data enhancement, especially in winter (top panel of Figure 2) where observa-
tions in some areas has doubled relative to V8 as shown in Figure 2. Additionally, the Amazon
region experienced significant data increase compared to V8, especially in JJA months as shown
in Figure 4.  The increase in retrieval yield over the Amazon region has been investigated in more
details by Deeter et al. (2021).

Here we focus on daytime data, and therefore there is a cut off at high northern latitudes in

the northern-hemisphere winter, and at high southern latitudes in the southern-hemisphere winter.
In general, high latitude regions (poleward of 65°) have strong seasonal variations in data cover-
age, with the northern high latitudes showing the highest coverage rates for both V9 and V8 in
June, July, and August, and the southern high latitudes exhibiting the highest rates in December,
January, and February as a result of less cloud in the summer. However, V9 successful retrievals
of spring (February, March, and April) and fall (September, October and November) seasons ex-
perienced a significant coverage gain in comparison to V8. Hence, the cloud detection scheme
modifications in the new V9 product resulted in an improvement in observational coverage, espe-
cially over land (Deeter et al., 2021).

**4.2 Analysis of standard and non-standard MOPITT product**
In this section, we present an analysis of Moderate Resolution Imaging Spectroradiometer
(MODIS) cloud heights and cloud mask products along with MOPITT retrieval cloud flag de-
scriptors to understand the impact of cloud conditions on the MOPITT observational coverage,
with a particular focus on observations over Canada.
CO TC were retrieved for a selected number of dates and locations by suppressing the cloud
detection scheme, so that all MOPITT L1 data were used to produce the L2 product regardless of
the cloud conditions. This non-cloud masked product will be referred to here as the non-standard
product. Analysis of the CO TC V8 L2 standard (cloud filtered) and non-standard product (non-
cloud masked) were performed for some selected cases. Figures 6a and 6b show the standard and
non-standard CO product on 16 August 2018, respectively, over the region between 78°W–92°W
and 44°N–60°N, which covers Ontario, Canada, near Hudson Bay. The standard (cloud masked)
product indicates that about 60% of the data are missing. Comparing it to the non-standard (non-
masked) product, some features can be observed in the non-standard product over the regions that
were missing data in the V8 standard product. A coherent structure is present between 50°N–54°N
(as it is indicated by pink and purple colors). The IASI TC for the same area and time was analyzed
to corroborate whether the features in the non-cloud-masked product are actual CO plumes
(Figures 6c). Comparing IASI CO TC on 16 August 2018 (Figures 6c) to the corresponding
MOPITT (Figure 6b) illustrate a strong CO plume around 50-55 °N and -94: -84 °W that is
apparent in both IASI and MOPITT. In the next section the MODIS cloud height product was used
to diagnose the cause of the missing (not retrieved in the V8 standard product) CO features.
**4.3 Regional analysis of MODIS cloud height and MOPITT data**

The MODIS swath (2330 km) is much wider than the MOPITT swath (640 km), so it
provides complete overlap for MOPITT passes. The MODIS cloud height (MOD6 L2) product
(Ackerman et al., 2008) has 5 km horizontal resolution at nadir (Ackerman et al., 1998). Therefore,
each MOPITT pixel can encompasses ~ 20 MODIS 1 x 1 km pixels. After co-location, relevant
MODIS cloud height values are gathered and averaged for each MOPITT pixel.

Figures 6d depicts the V8 MOPITT cloud index (see Table 1), for the case on 16 August
2018. Retrievals were assigned cloud index 2 (MODIS and MOPITT clear, grey color), 3 (MODIS
clear and MOPITT cloudy, dark blue), and 4 (low clouds, cyan color). Figures 6d shows that the
V8 L2 data on 16 August 2018 case were retrieved based on clear and low cloud conditions as
indicated by flag number 2 and flag number 4. Figure 6e displays the MODIS cloud height (and
cloud mask for the same swath on 16 August 2018. Comparing the low cloud retrieval area (cyan
color) to the corresponding MODIS cloud height (Fig. 6e) and cloud mask (Figure 6f), it can be
seen that this area has cloud percent (the term "cloud" encompasses water clouds and aerosols) by
more than 90% and has cloud heights less than 1 km, as illustrated by the grey color (Figure 6e).
The MODIS cloud height also shows other areas that have low clouds (grey and blue colors) where
there were no retrievals in the V8 standard product. Those pixels collocate with the coherent
pattern region (between 52°N–54°N) that were shown in the non-masked product (Figure 6b).
Therefore, it appears that some of the potential retrievals were missed in the V8 standard retrieval
due to misidentification of low cloud pixels. It is necessary to examine additional cases using the
same approach to determine whether these findings are widespread.

**4.4   Analysis of V8 cases under different cloud and pollution conditions**
In this section, additional cases are investigated by analyzing the cloud filtered (V8 standard)
and the non-cloud masked, along with the MODIS cloud height and cloud mask products. Figure
7 shows the results over Canada, on 12 April 2010 and it indicates that, about 70% of the data are
missing in the standard retrievals (Figure 7a). However, the non-cloud masked product (Figure 7b)
captures notable features between 54°N–56°N and 90°W–98°W (as indicated by the red colors in
Figure 7b). The MOPITT cloud flag description on 12 April 2010 (Figure 7d) reveals that all L2
data were retrieved under clear conditions (MODIS cloud percent less than 5%) as indicated by
the flag number 2 (grey color) and the MOPITT diagnostics data (Figure 7c). However, the
corresponding MODIS cloud height (Figure 7e) showed an area of very low cloud heights that are
less than 500 m (around 54°N–56°N), where the MOPITT measurements were not retrieved
completely in the V8 standard product as they were considered cloudy (with more than 5% cloud
cover, see Figure 7f). Comparing this area to the collocated non-masked CO product (Figure 7b),
it can be noted that it exactly matches the coherent pattern that was observed between 54°N–56°N.
Looking to IASI CO TC for the same time and location on 12 April 2010 (Figures 7c), it can be
seen that most of the CO features in the area of 52-56 latitude and -100: -92 longitudes (Figure 7b)
are not captured as well due to their cloud detection scheme.
An unusually active forest fire season occurred in the vicinity of Fort McMurray, Alberta, in
May 2016. Figures 8a and 8b display the V8 standard and non-standard CO TC on 6 May (day),
respectively. Again, the non-standard CO product exhibits a notable coherent pattern over some
areas that were not retrieved in the standard product. On 6 May 2016, there is a CO plume around
50°N–52°N and 108°W–112°W longitude that is indicated by the purple colors (Figure 8b) and it
is completely missed in the V8 standard product. On the other hand, IASI shows a consistency
with the non-masked MOPITT product where a prominent CO plumes was observed around 50-
56 latitude and -112: -104 longitudes which coincide the corresponding MOPITT (Figure 8b).
The elevated CO values on 6 May 2016 is likely to be a result of Fort McMurray fire
emissions in northern Alberta (as indicated by MODIS fire images, not shown). Considering the
low cloud detection during the Fort McMurray fires, the MODIS cloud height data of the
corresponding MOPITT pixels on 6 May 2016 (Figure 8e) suggest that none of the low cloud (blue
colors) pixels were retrieved in the standard product as it is implied by the MOPITT flag number
(Figure 8d) (all values are 2).

**4.5   MODIS height comparison with MOPITT radiance ratio**

As it is mentioned in section 3 the MOPITT retrieval algorithm only retrieve CO in clear-
sky conditions. The cloud detection scheme utilizes information from both MODIS cloud mask
product and the MOPITT's thermal-channel radiances (Warner et al., 2001). Radiance from the
MOPITT 4.7 μm thermal channel radiance is compared to the calculated model for each pixel. If
the measured/calculated radiance ratio is greater than the threshold (which is one for V8), then the
observation is considered "clear".
In MOPITT V8 and before, the MODIS test value supersedes the MOPITT test value over
land, i.e., the MOPITT pixel will be considered "clear" if the MODIS test is "clear" and the
MOPITT test is "cloudy". However, the MOPITT pixel will be considered "cloudy" if the
MOPITT test identifies the pixel as clear and the MODIS test identifies the pixel as cloudy. Hence
V8 level 2 retrievals are processed over land just if the MODIS test passes. For the MOPITT V9
product, Deeter et al., (2021) modified the cloud detection algorithm by allowing CO retrievals
when the MOPITT radiance ratio (MRT) test indicates the pixel is clear although the MODIS cloud
mask test assigns the pixel as cloudy. Deeter et al., (2021) modified the cloud detection algorithm
by allowing CO retrievals when the MOPITT radiance ratio test indicates the pixel is clear alt-
hough the MODIS cloud mask test assigns the pixel as cloudy.
To understand how the new V9 cloud detection scheme improved the coverage rate, an anal-
ysis of the MRT and MODIS cloud height has been conducted for many cases over Canada. The
data on 6 May 2016 are presented here as a case study and are shown in Figure 9. It can be seen
that there is a negative correlation between the MRT and MODIS cloud height with a slope of -
0.06 and a correlation of R = 0.68.
A Box and Whisker plot of MRT and the corresponding MODIS cloud heights for various
groups are displayed in Figure 9b. Since the modified cloud detection scheme of V9 relies on the
MRT threshold test (the threshold value is one), it is expected that, most of the observations with
cloud heights up to 3 km are incorporated in V9 retrievals as illustrated Figure 9b.
Figures 9c and d in the bottom panel depict the histograms density of MODIS cloud heights of
the corresponding MOPITT clear/MODIS clear observations and MOPITT clear/MODIS cloudy,
respectively on 6 May 2016. The successful retrievals using MOPITT clear/MODIS clear pixels
and MOPITT clear/MODIS cloudy are 45.4%, and 14.2%, respectively.
Since MRT correlates negatively with the low cloud heights (as indicated above in Figure 9a),
the low cloud cases are included in V9 (Figure 9d) with high proportion of heights less than 3 km.
Hence, adding low cloud observations as a result of considering MRT values of greater than 1
enhances the MOPITT coverage percentage by 14.2% compared to 45.4% successful retrievals
without considering the low cloud cases. The total coverage rate is about 60% with about a 30%
(14.2/45.4) gain in data coverage. Therefore, using the MRT cloud test independently in V9 cloud
detection scheme resolved the problem of low cloud miss-detection over land, which results in a
significant data coverage increase, especially over the region of Canada.

## 4.6  MOPITT and IASI comparison
We examine the impact of the increased observational coverage in the MOPITT TIR V9
product in comparison to IASI data over Canada for three case studies. The first and third cases
are associated with biomass burning emissions (6 May 2016 and 16 August 2018), while the se-
cond case represents typical conditions with no extreme air pollution (12 April 2010). Figure 10
shows maps of 1-day/morning overpasses of CO total columns measured by MOPITT and IASI
on 6 May 2016. Figures 10a and 10b show MOPITT V9 and V8 data, respectively, while Figures
10c and 10d show the corresponding IASI data (collocated with MOPITT) and the entire IASI CO
field (gridded in 0.25° x 0.25° bins), respectively. As seen in Figures 10a and 10b, there is a gap
with missing MOPITT data in V8 that extends across Alberta and Saskatchewan between 110°W
and 100°W, but the data are present in V9 as a result of the improved retrievals in low cloud
conditions (as indicated by the MODIS cloud heights in Figure 8e). The high CO total column
values that are added in MOPITT V9 product coincide with the high AOD and OMI UV Aerosol
Index (UVAI) values (not shown) from the Fort McMurray fire emissions. Smoke was transported
from the eastern part of Alberta, moving into Saskatchewan and central Alberta in the vicinity of
the high CO values. Interestingly, the added retrievals in V9 exhibit a pattern that is consistent
with the IASI data (Figure 10c). Since IASI has daily global coverage compared to MOPITT's 3-
day global coverage, the entire smoke plume is captured by IASI (Figure 10d).
Figure 11 depicts the scatter plots of IASI and MOPITT TIR V9 and V8 retrievals over
Canada on 6 May 2016 and for the entire month of May 2016 (monthly average). IASI and
MOPITT data are gridded in 0.25x0.25 deg., then the daily collocated data are selected for the
analysis.
In general, IASI and MOPITT retrievals are consistent to a large extent with a correlation
coefficient of 0.98-0.99 and 0.97-0.98 for V8 and V9, respectively. However, IASI has higher
values than MOPITT over Canada, with the slope varying from 1.04 to 1.06. Total CO column
biases for V9 are somewhat larger than for V8 products; with a slope for V9 of 1.05 and 1.06,
whereas for V8 it is 1.04 and 1.05 for data on 6 May 2016 and for all of May 2016, respectively.
These discrepancies occur at high CO values, and since the added data in V9 are mainly in heavily
polluted regions, the IASI bias is greater for V9 than V8.
For the second case analysis on 12 April 2010, V9 (shown in Figure 12a) exhibited greater
data coverage relative to V8 (Figure 12b) around 126°W, 56-60°N, 90°W, 56-60°N, and 80°W,
44°N as a result of retrievals of low cloud height pixels (Figure 7e). Figure 12c shows generally
good agreement between IASI CO total column values and corresponding MOPITT retrievals. As
this time of year has no extreme air pollution sources (such as forest fire emissions), the CO total
column values over land are in the range of 20-30 $10^{17}$ molecules/cm$^2$, which can be seen in the
whole IASI CO total column field in Figure 12d. Consequently, as shown in Figure 13, the IASI
biases with MOPITT V8 and V9 are generally similar, with comparable correlations and slopes.
Comparison of MOPITT V9 and V8 on 16 August 2018 over Canada, in Figure 14, shows the
greater number of successful MOPITT retrievals in V9 that were discarded in V8. The added data
in V9 are around 80°W-90°W, and 50°N-56°N and 100°W-117°W, 54°N-56°N. Those regions are
associated with cloudy areas of relatively low cloud heights as indicated by the MODIS cloud
mask and height (Figure 6e and 6f). As shown in Figure 14c, the IASI observational pattern is
generally consistent with the corresponding MOPITT CO total columns. However, there is an ap-
parent positive IASI bias around 80°W-90°W and 54°N where IASI CO values exceed 50 x$10^{17}$
molecules/cm$^2$ compared to 30 x $10^{17}$ molecules/cm$^2$ for MOPITT. These high CO values are as-
sociated with the dense pollution plume that extends across Canada as shown in the map in Figure
14d with the whole IASI observational scene and in Figure 15a with the MODIS Terra image
overlaid with the thermal anomaly spots.
The scatterplots of CO total column values between IASI and MOPITT TIR V9 and V8 for
August 2018 are shown in Figure 15b. The slopes of the relationship between IASI and MOPITT
V8 data on 16 August 2016 and for all of August are 1.09 and 1.07, respectively. Since the added
MOPITT retrievals in V9 are associated with higher CO total column values, the slopes increase
to 1.12 and 1.1, respectively, with smaller correlation coefficients. CALIPSO total attenuated
backscatter at 532 nm on 16 Aug. 2018 for the two yellow swaths shown in Figure 15a are pre-
sented in Figures 15c and 15d. The smoke aerosols were observed at altitudes between 2 km and
6 km, as measured by CALIPSO, indicating that convective lofting may have elevated the fire
emissions above the boundary layer into the free troposphere. The large CO enhancements ob-
served by IASI around -80°W-90°W and 54°N (Figure 14c) are collocated with the maximum
aerosol backscatter coefficient at ~3 km detected by the CALIPSO lidar (Figure 15d). However,
CO in this area is underestimated by MOPITT TIR relative to IASI (Figure 14b) resulting in 12%
overall bias over Canada on 16 August 2018 (as indicated by the slope of 1.12).
Similar results were found by Turquety et al. (2009) as their study revealed that IASI CO is on
average 35% higher than MOPITT in regions of elevated CO concentrations during extreme fire
events. There are many factors that could explain the discrepancies between IASI and MOPITT
during pollution events. One of them is the different horizontal resolution of the two instruments
(22 km × 22 km for MOPITT and a 12-km diameter for IASI), especially above inhomogeneous
scenes. A second major factor that could contribute to the differences between the MOPITT and
IASI retrievals is the a priori used in the retrievals. IASI uses a fixed a priori while MOPITT has
variable a priori profiles. George et al. (2015) examined the impact of the a priori on the IASI and
MOPITT data and found that using the same a priori constraints slightly improved the correlation
between the two data sets and reduced the large discrepancies (total column biases over 15 %)
observed at some places by a factor of 2 to 2.5. However, other regions did not show any bias
reduction. A third factor is the difference in vertical sensitivity between the two instruments, as
reflected by their averaging kernel matrices (the sensitivity of the retrieval to the abundance of CO
at different altitudes). The instruments have different degrees of freedom for signal (DOFS), which
is given by the trace of the averaging kernel matrix; the DOFS of the MOPITT retrievals is lower
than the corresponding IASI retrievals (not shown). Although both instruments in general have
good sensitivity in the middle troposphere, IASI's averaging kernel indicating greater sensitivity
in the upper troposphere as well. The difference in averaging kernels for the instruments can be
attributed to instrumental and retrieval factors (George et al., 2015). For example, surface emis-
sivity and water vapor are treated differently in the two retrieval algorithms. The MOPITT algo-
rithm retrieves emissivity simultaneously with CO but uses a fixed water vapor profile from
NOAA/National Centers for Environmental Prediction (NCEP), while IASI assumes a fixed emis-
sivity but estimates the water vapor amount (Barré, J., et al., 2015). Understanding how the factors
discussed here, as well as others, potentially contribute to the discrepancies between MOPITT and
IASI will be further investigated in future work

## 5. Conclusion

In this study, an analysis has been performed to understand the improvements in observational coverage over Canada in the new MOPITT V9 product. Temporal and spatial analysis of V9 indicates a general coverage gain of 15-20% relative to V8 which vary regionally and seasonally. For example, the number of successful MOPITT retrievals in V9 was doubled over Canada in winter.

The standard (cloud filtered) V8 CO TC (L2) product was compared with a non-standard (non-cloud masked) version of the retrievals for selected days to understand the observation gain in V9 relative to V8. The results reveal some coherent structures of CO plumes that were observed frequently in the non-cloud masked product but which were missing in the standard the V8 product. Those features are not captured in V8 standard product because the cloud detection scheme did not properly detect many low cloud cases over land.

The modified V9 cloud detection scheme utilizes MRT test (threshold value of 1) individually which allow CO retrievals when the MRT test indicates the pixel is clear although the MODIS cloud mask test assigns the pixel as cloudy. Since MRT correlates negatively with the low cloud heights, most of low cloud observations (up to 3 km) are included in V9 L2 retrievals. Hence, the incorporation of the MRT test over land will resolve the low cloud detection issue as it is demonstrated by MODIS cloud height correlation. Hence, adding low cloud observations as a result of considering MRT values of greater than 1 enhances the MOPITT coverage percentage.

The improved V9 cloud detection scheme benefits regions that are often characterized by high aerosol concentrations (e.g. biomass burning emissions). An analysis of MOPITT and IASI CO are conducted for three cases. The first and third cases are associated with biomass burning emissions, while the second case represents typical conditions with no extreme air pollution. The added retrievals in V9 exhibit a pattern that is generally consistent with the corresponding IASI data. In general, the IASI MOPITT comparison indicated discrepancies at high CO values. Since the added data in V9 are mainly characterized by high aerosol load (low cloud cases) that are usually associated with high CO values as a result of fire emissions, the IASI bias is greater for V9 than V8. So, IASI MOPITT CO TC comparison indicated generally good agreement with about 5-10% positive bias which increases in highly polluted scenes.

## ACKNOWLEDGMENTS

The authors would like to thank the CSA (Canadian Space Agency) for their financial
support of this research. NCAR (National Center for Atmospheric Research) is sponsored by the
National Science Foundation and operated by the University Corporation for Atmospheric
Research. The NCAR MOPITT project is supported by the National Aeronautics and Space
Administration (NASA) Earth Observing System (EOS) Program. The MOPITT team
acknowledges support from the Canadian Space Agency (CSA), the Natural Sciences and
Engineering Research Council (NSERC) and Environment Canada, and the contributions of
COMDEV (the prime contractor) and ABB BOMEM. The authors thank the AERIS infrastructure
(http://www.aeris-data.fr) for providing access to the IASI CO data.

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

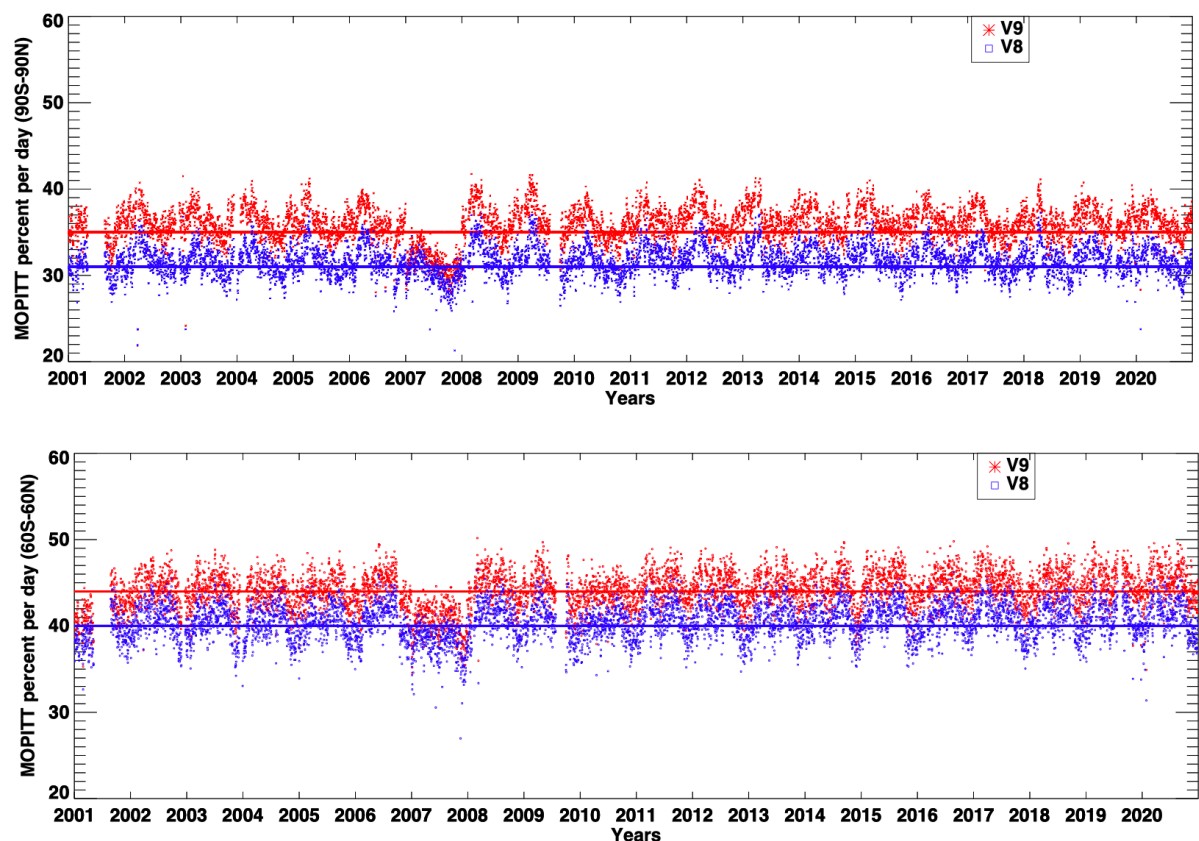


Figure (1) The percentage of successful daily MOPITT retrievals between 90°S–90°N and 60°S–
60°N from 2000 to 2020 for V9 and V8. The solid lines represent the average successful retrieval
for the entire period.

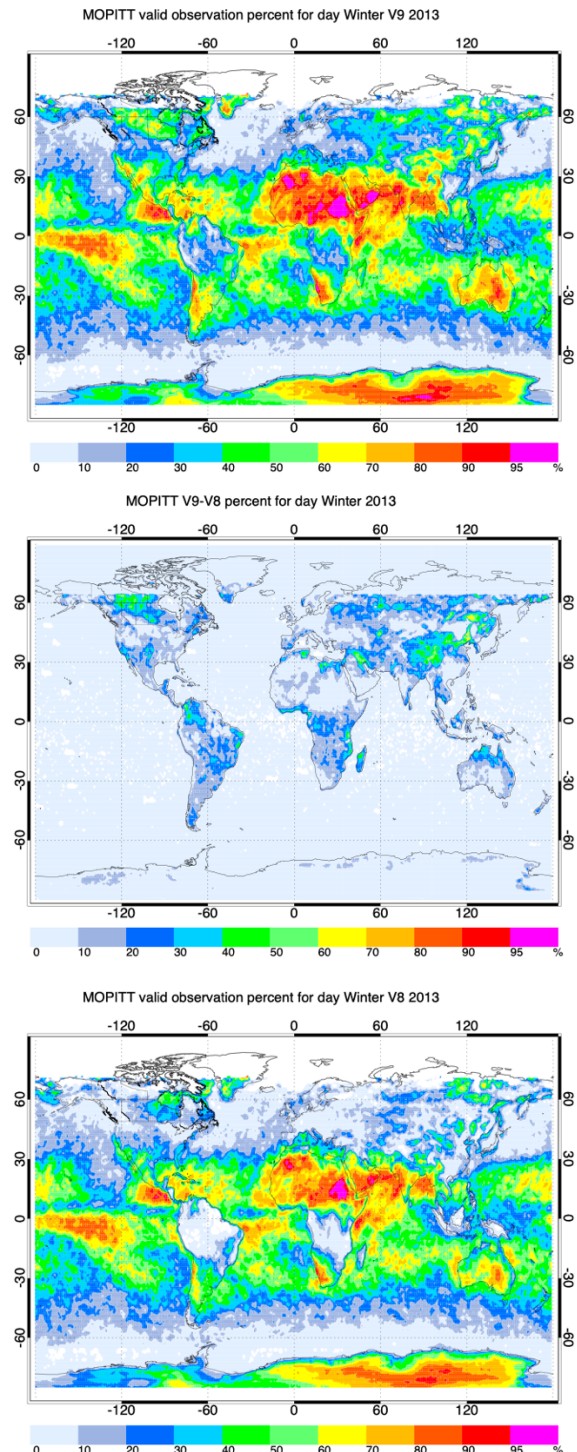


Figure (2) Seasonally averaged spatial distribution of the successful MOPITT retrievals in winter
2013 for V9 (top panel), V8 (bottom panel) and V9-V8 (middle panel). Data were aggregated
into 1° × 1° bins.

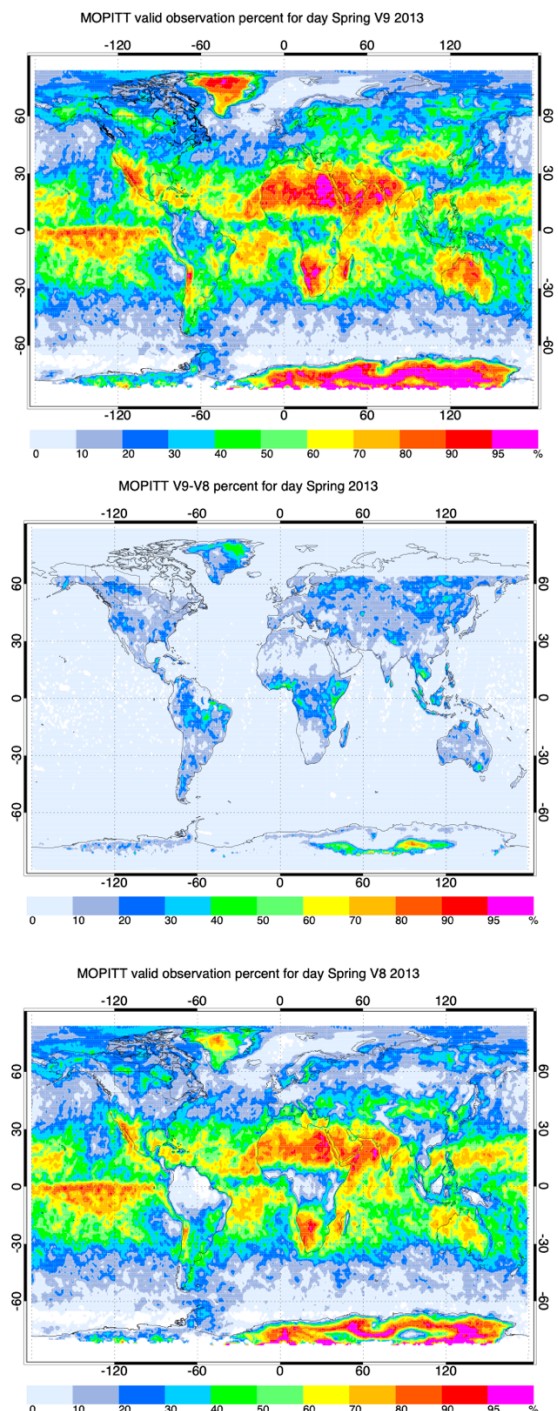


Figure (3) The same as Figure 2 but for spring season.

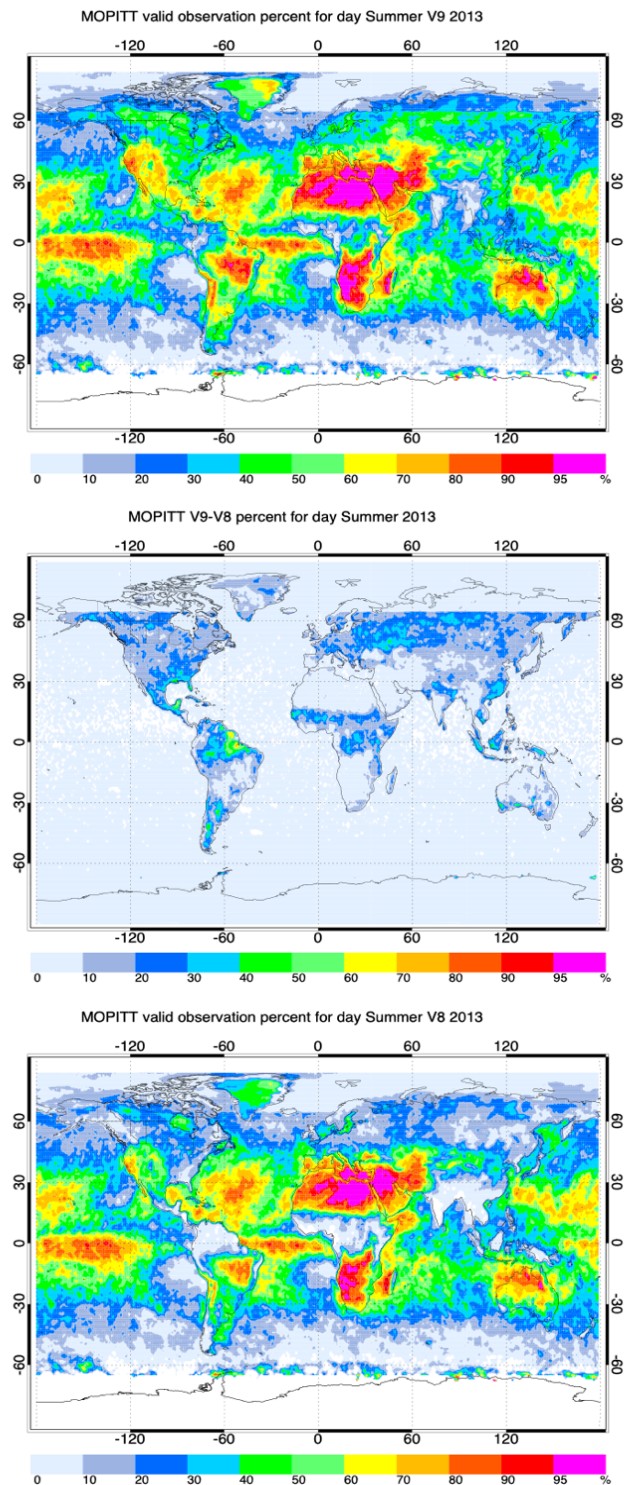


Figure (4) The same as Figure 2 but for summer season.

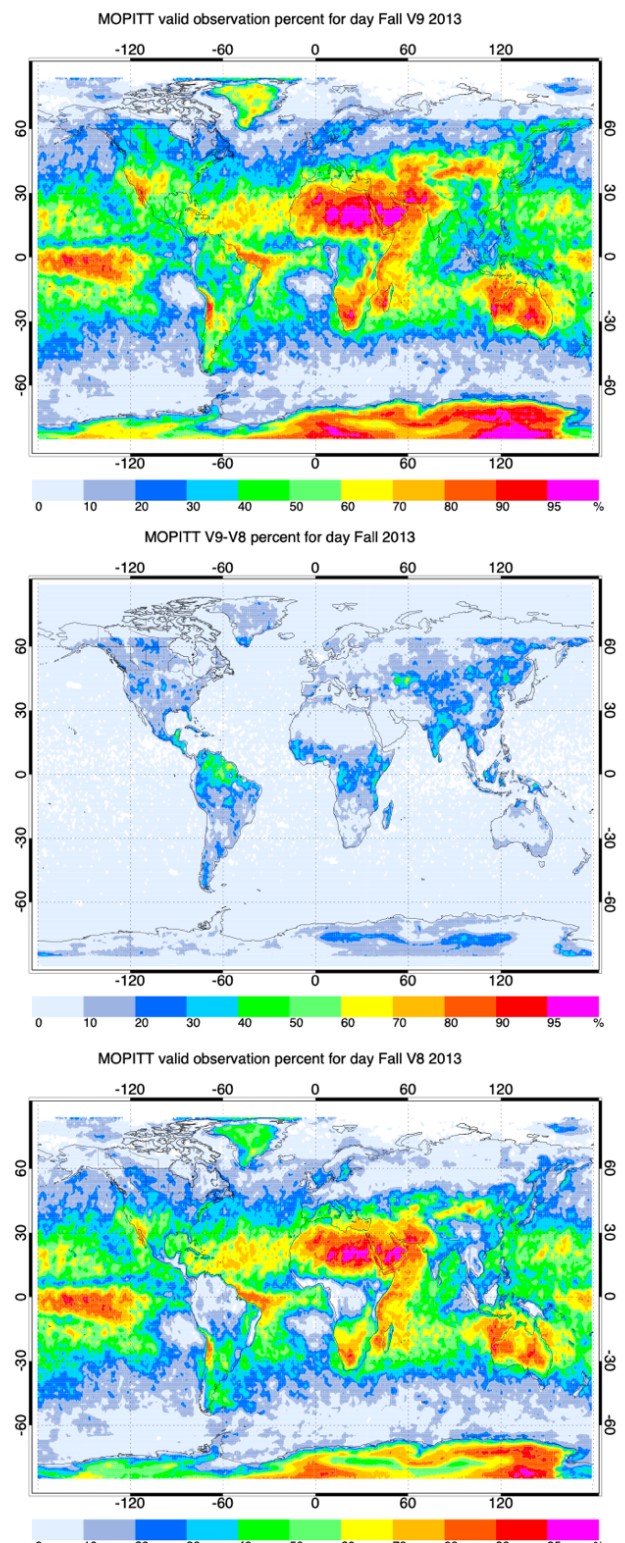


Figure (5) The same as Figure 2 but for fall season.

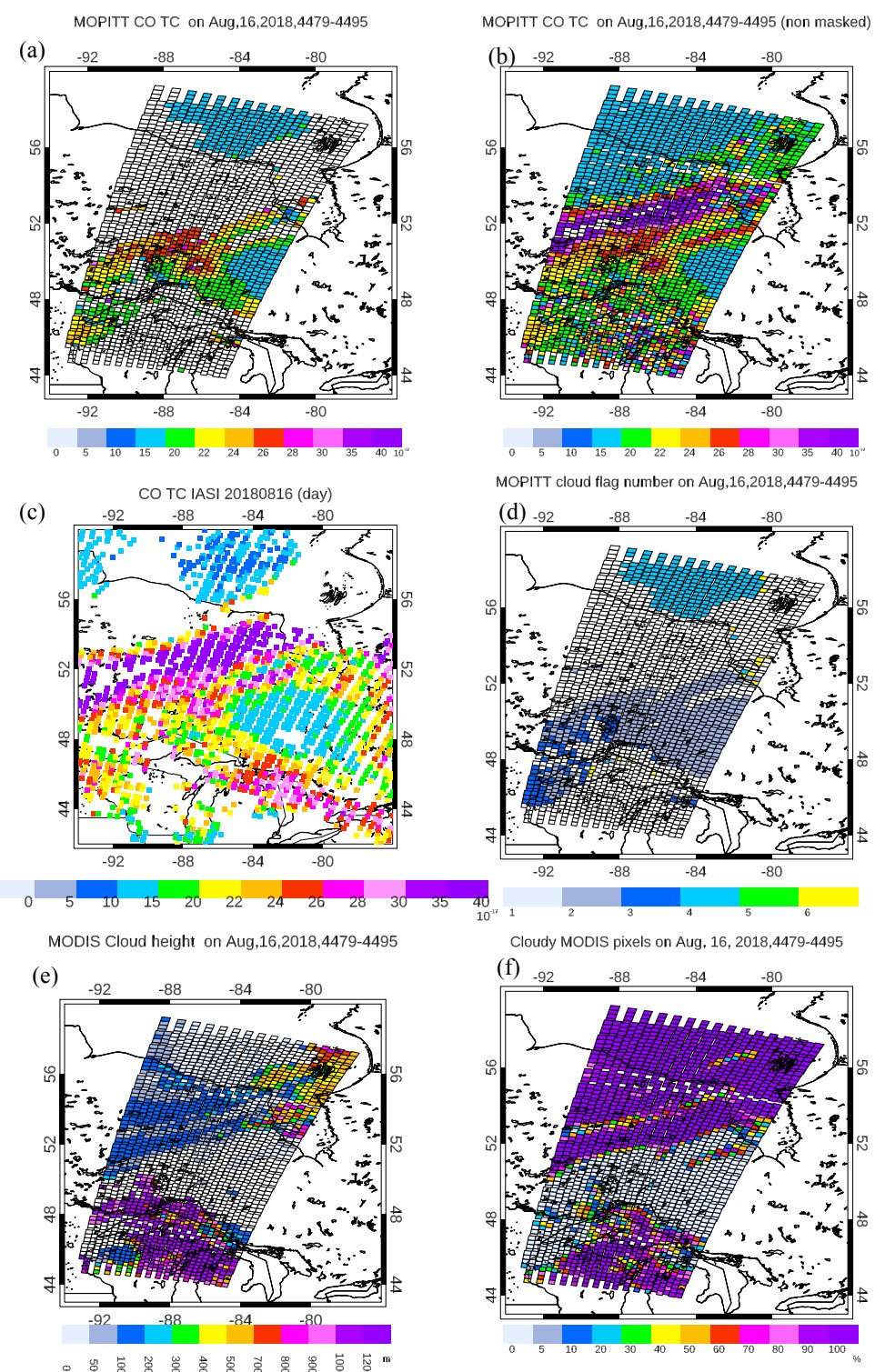

Figure 6. (a) Standard (cloud masked), (b) non-standard (non-cloud masked) CO TC, (c) IASI CO
TC, (d) MOPITT cloud flag number, (e) MODIS cloud height, and (f) cloud mask on 16 August,
2018. The faint black squares represent MOPITT pixels (22 km x 22 km) for all L1 observations.

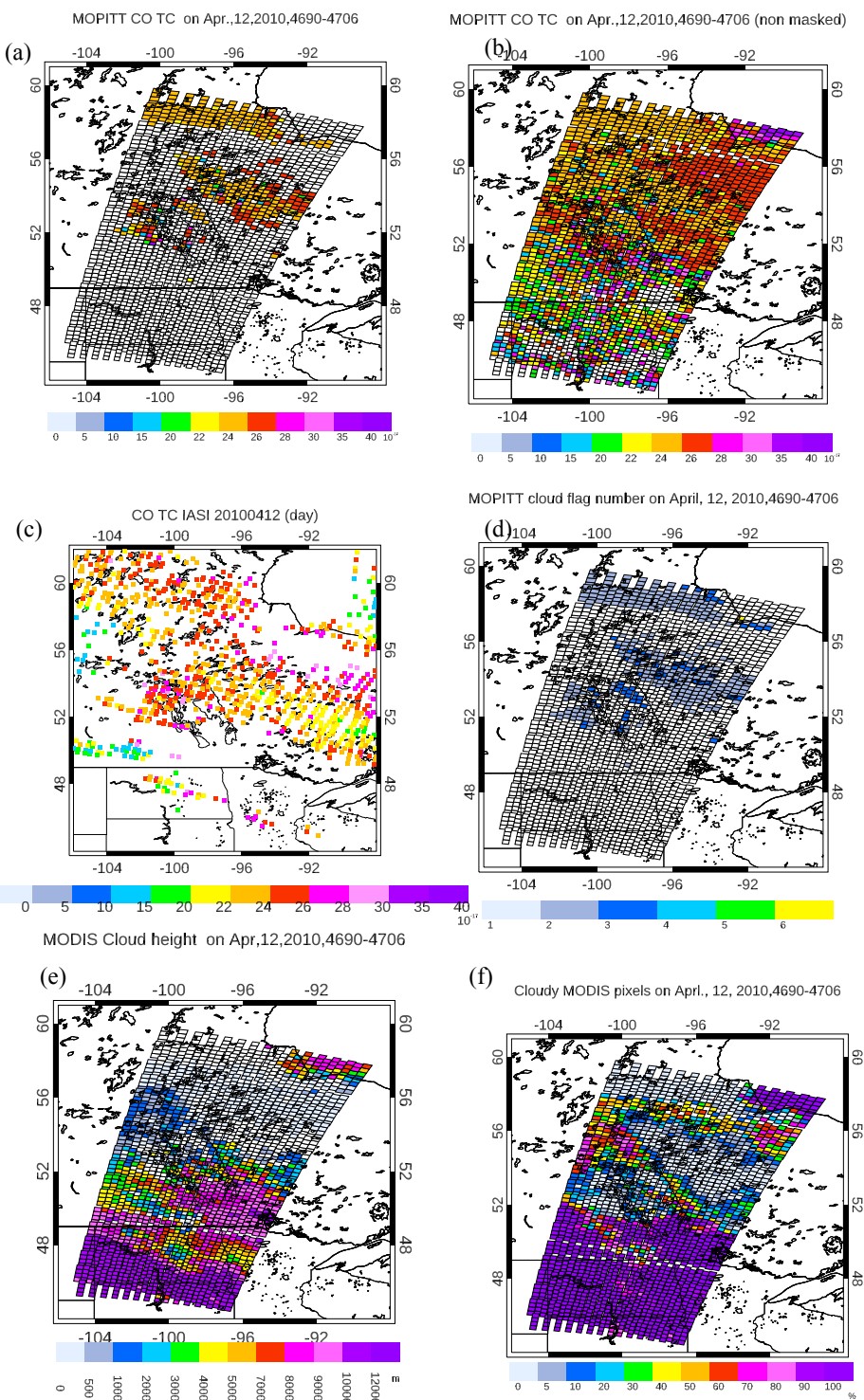

Figure (7) The same as Figure 3, but for 12 April 2010.

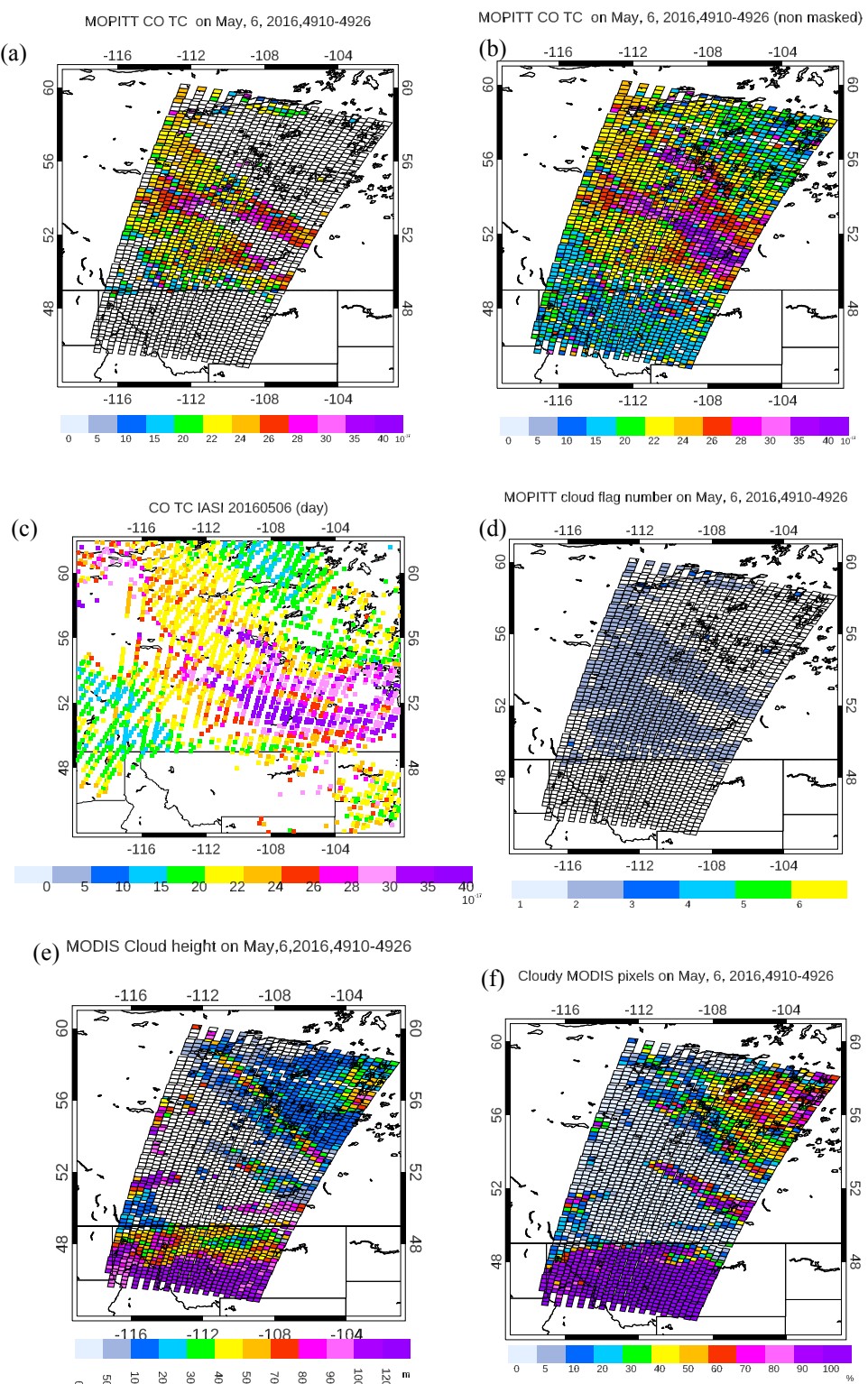

Figure (8) The same as Figure 3, but for 6 May 2016.


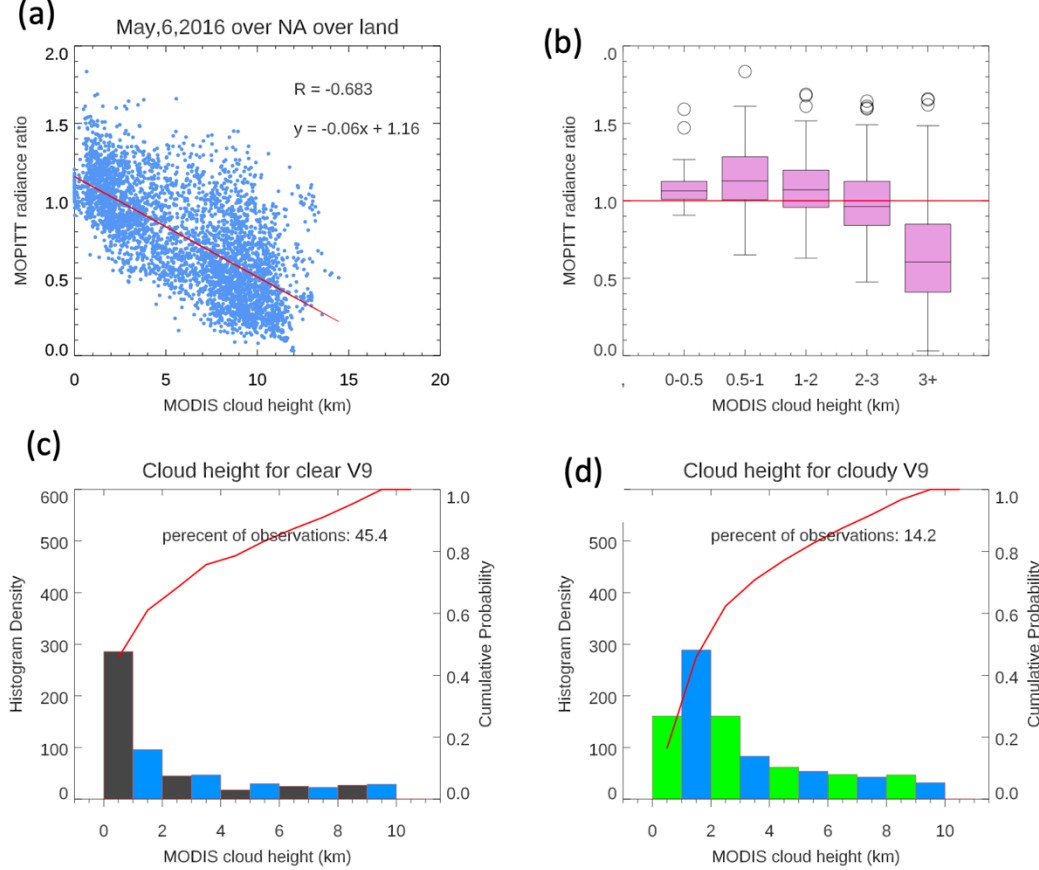


Figure (9) (a) scatter plot correlation between MOPITT radiance ratio (MRT) and MODIS cloud
height, (b) A *Box* and *Whisker plot* of MRT and various MODIS cloud height groups, (c) The
histogram density of MODIS cloud heights of MOPITT clear/MODIS clear observations, (d) The
histogram density of MODIS cloud heights of MOPITT clear/MODIS cloudy observations on 6
May 2016.

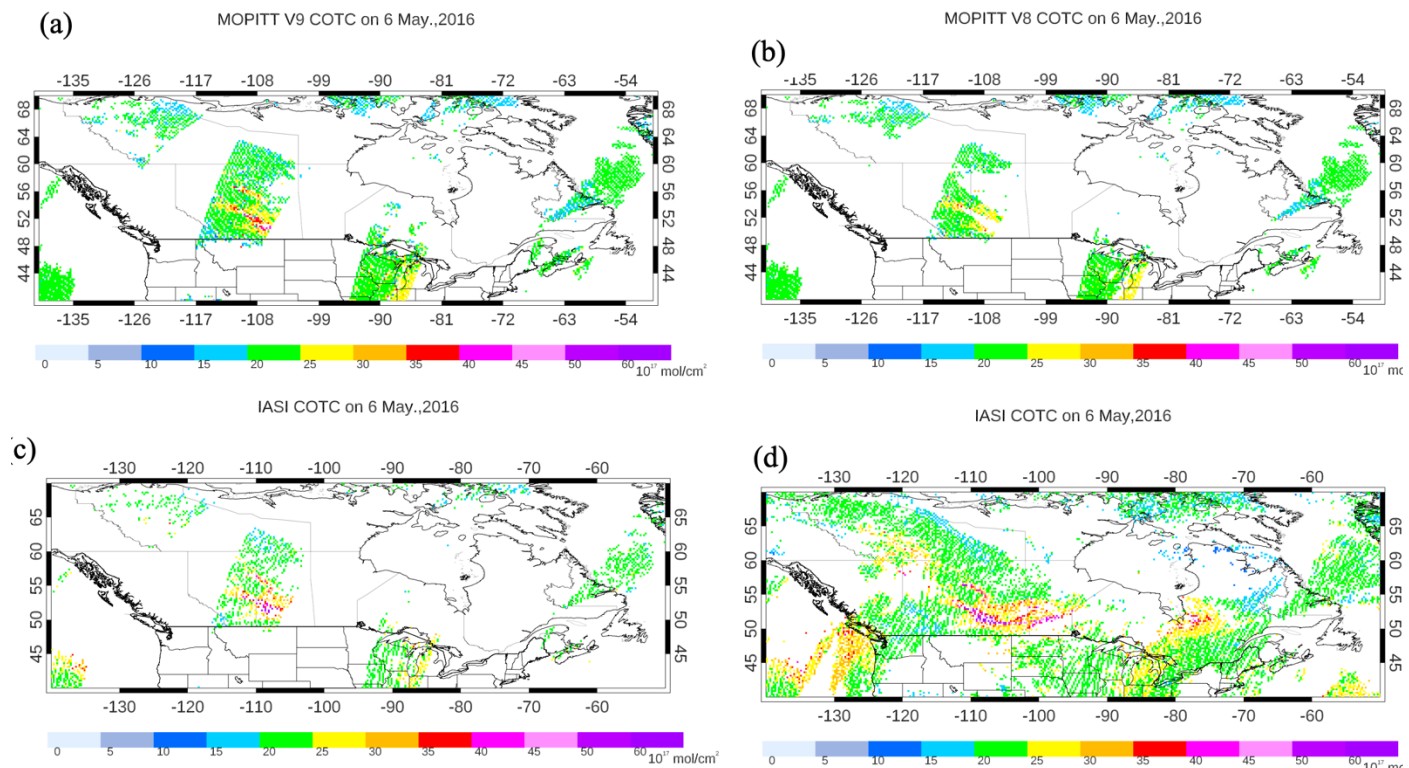


Figure (10) MOPITT CO total column for V9 (a) and V8 (b). IASI CO total column observations
of the corresponding with MOPITT (c) and the entire IASI CO retrievals (d) on 6 May 2016.

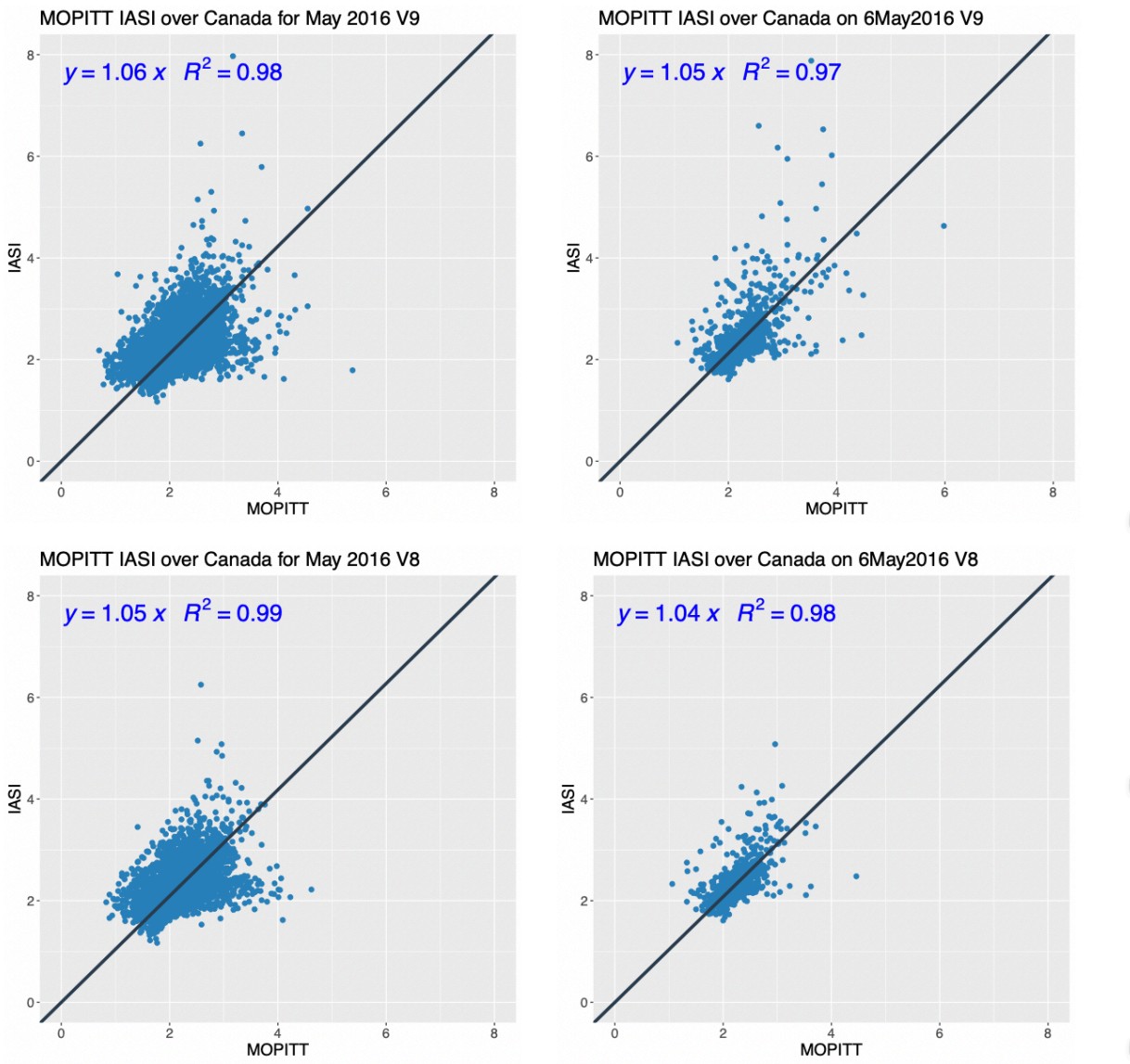


Figure (11) Scatter plots of the IASI and MOPITT CO retrievals in $10^{18}$ molecules/cm$^2$, for 6 May
2016 and the monthly averaged May 2016. The correlation coefficient and the regression slope are
reported.

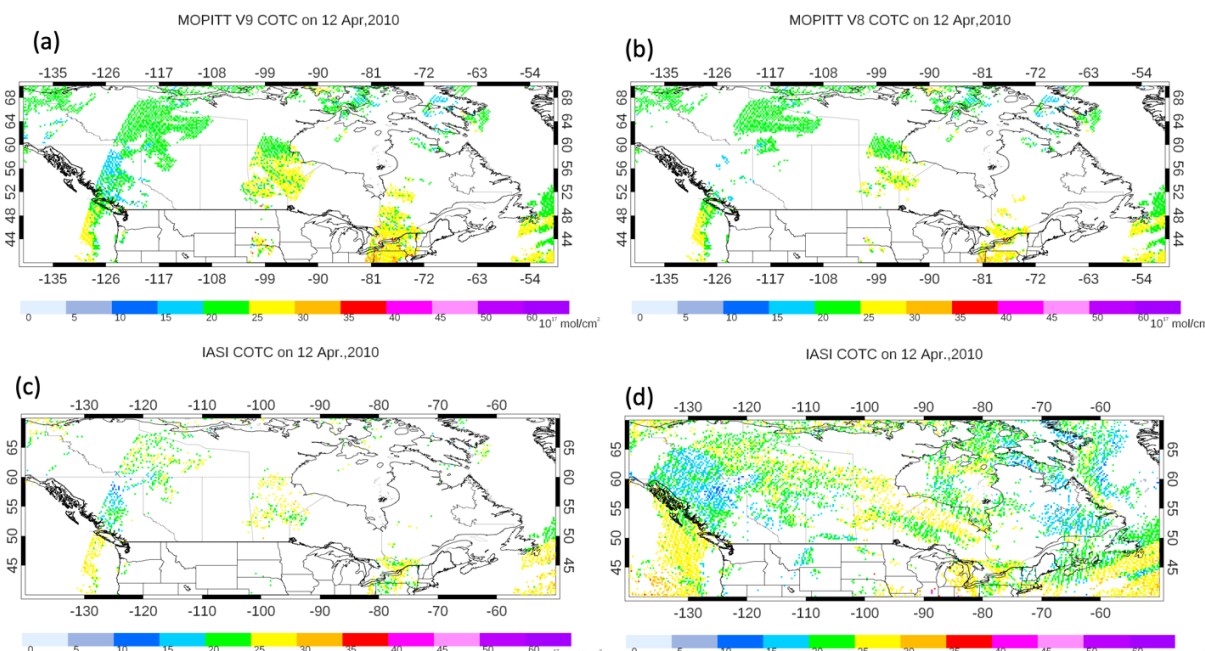


Figure (12) The same as Figure 10, but for 12 April 2010.

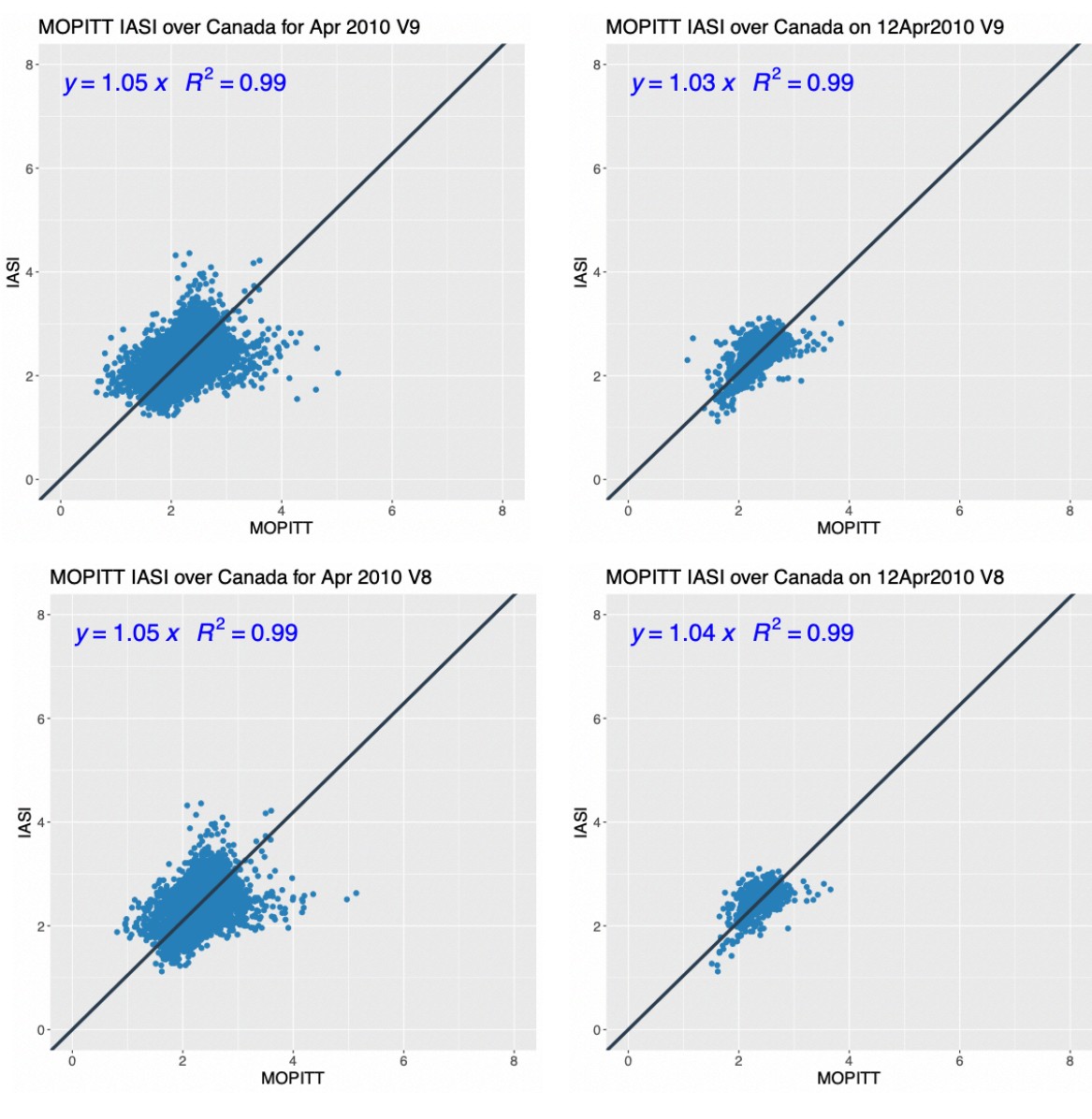



Figure (13) The same as Figure 11, but for 12 April 2010.

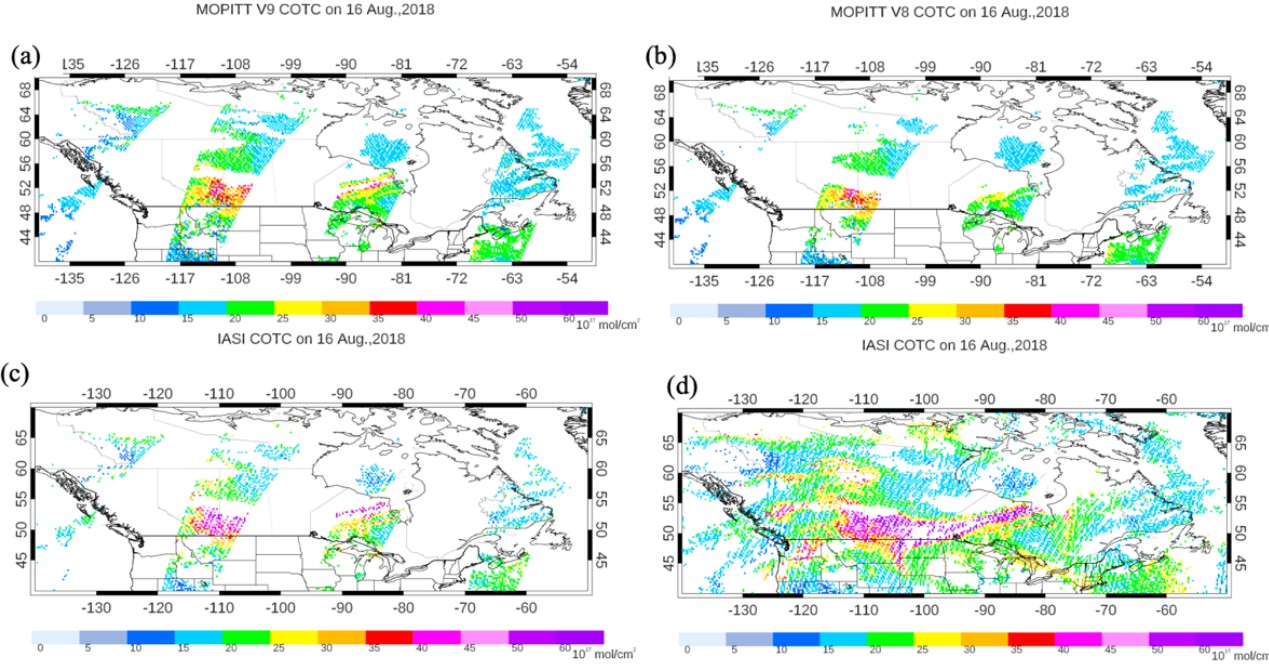


Figure (14) The same as Figure 10, but for 16 August 2018.

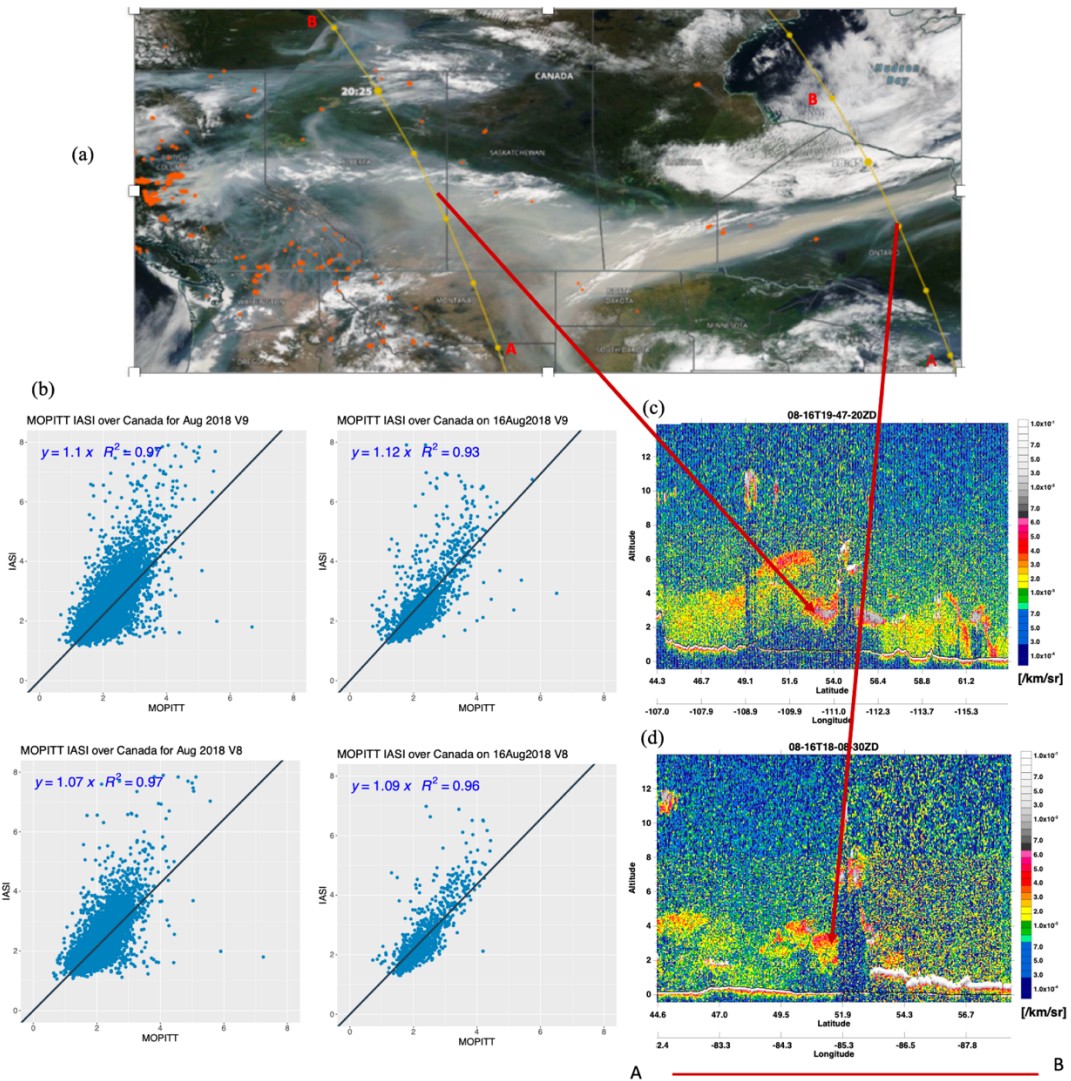


Figure (15) (a) MODIS Terra overlaid with fire points (red points), (b) scatter plots between IASI
and MOPITT TIR V9 and V8 and (c-d) daytime CALIPSO 532nm total attenuated backscatter on
16 August 2018.