# Peer review of "Analysis of improvements in MOPITT observational coverage over Canada"

_Atmospheric Measurement Techniques, 2021_

## Author Comment (AC1)

We would like to thank the reviewer for his/her positive and insightful comments on the manuscript. Below is our response (black) to the referee comments (red).

**sRC1**: 'Comment on amt-2021-112', Anonymous Referee #1

Still I cannot recommend the manuscript in its present form for publication because it seems that the authors have stopped half-way in their analysis:

The main goal of the paper that was submitted in June 2021 was not to propose a new MOPITT product. It was to investigate the issue of increasing the number of MOPITT observations and suggest a way of increasing the coverage rate in cloudy conditions. Work was being done in parallel with our colleagues at NCAR to produce a new product, version 9 (V9), and a description of this new product was published recently in Remote Sensing of Environment by Deeter et al., 2021. This new MOPITT V9 version product is available now to the public. As a result, we have restructured the paper to better reflect this and to complement Deeter et al. (2021). We have also changed the title to better describe the unique contribution of this analysis to the evaluation of the MOPITT V9 product. In addition, this study shows comparisons with the IASI instrument that are not presented elsewhere.

(1) the cases are only presented as a set of images which makes it very difficult for the reader to infer quantitative conclusions,

We previously selected a few case studies to better illustrate the issues with the observational coverage. In the revised manuscript, we have reduced the number of cases and have presented a more quantitative analysis of the impact of clouds on the observational coverage as new sections 4.5 and 4.6 are added.

(2) chi2 is tentatively proposed at the end as a quality index measure, however, without any quantitative analysis or explanation;

This was presented as a possible approach. However, since the modified MOPITT cloud detection algorithm of V9 does not use MODIS cloud height, instead it uses the MRT threshold method, we decided to remove the chi2 section from the revised paper.

(3) I would have expected that a new enhanced dataset is proposed and presented and at least a first quantitative comparison with external data (e.g. IASI) in comparison with the 'old' dataset is provided. However, this is not the case and the reader remains with the impression that one can get more from MOPITT by also cloud-affected scenes, but it remains open what is the final new dataset and how large might the related uncertainties be.

We apologize for the confusion, but as we noted above, the main goal of the paper was not to propose a new MOPITT dataset. A new dataset was produced in parallel at NCAR and is now available to the public. The revised MOPITT cloud detection algorithm used in this new dataset is discussed by Deeter et al. (2021). Hence, we have revised the paper to complement Deeter's work by providing a detailed analysis of the improvements in V9 MOPITT observational coverage. The analysis is conducted to understand the impact of cloud conditions on the MOPITT observational coverage of V9 and V8, with a particular focus on observations over Canada using Moderate Resolution Imaging Spectroradiometer (MODIS) cloud heights and cloud mask products along with MOPITT retrieval cloud flag descriptors.

Then a quantitative comparison of MOPITT V9 TIR with the corresponding IASI CO is conducted for three cases. The first and third cases are associated with biomass burning emissions, while the second case represents typical conditions with no extreme air pollution. The study revealed positive bias of IASI relative to MOPITT which mostly occur at high CO values, and since the added data in V9 are mainly in heavily polluted regions, the IASI bias is greater for V9 than V8. Understanding the factors that potentially contribute to the discrepancies between MOPITT and IASI will be further investigated in future work. Additionally, the performance of the revised cloud detection algorithm is evaluated through validation based on a set of in-situ CO

profiles acquired during the ACRIDICON-CHUVA campaign and NOAA Aircraft Profiles by Deeter et al., 2021. Given the emphasis of the analysis on the impact of clouds on observational coverage, expanding the analysis to include TROPOMI would require a more challenging validation focus of the manuscript because of the differences in timing and spatial resolution of the MOPITT and TROPOMI measurements (morning, 10.30 am, compared to afternoon, 1.30 pm, and 22x22 km compared to 4x5 km).

Specific comments:

L71-75:

Could you be a bit more specific what the effect of clouds separately on the two CO retrievals (TIR and NIR) are?

L109:

Why are only the TIR CO products used here? It may be instructive to compare TIR and NIR retrievals with and without the cloud mask.

We did not mean to investigate the effect of clouds on the retrievals, the aim of the manuscript is to discuss the performance of the MOPITT cloud detection scheme in low cloud conditions. Since the MOPITT cloud detection algorithm is the same for TIR and NIR, the MOPITT coverage rate (L2 successful retrievals) is the same for both channels and hence there is no need to use NIR channel.

L108, 117:

Please specify in the instrument description the swath width, pixel size etc. which is referred to later on.

Thanks for the comment, it is added in lines 110:113 in the revised manuscript.

L245:

How is the MODIS cloud height for a MOPITT pixel determined (should also be described in the instrument-section)?

Thanks for the comment, it is added in lines 259:264 in the revised manuscript.

Since the MODIS swath (2330 km) is much wider than the MOPITT swath (640 km), it provides complete overlap for the MOPITT passes. The MODIS cloud height (MOD6 L2) product (Ackerman et al., 2008) has 5 km horizontal resolution at nadir (Ackerman *et al.*, 1998). Therefore, each MOPITT pixel can encompass approximately 20 MODIS 5 x 5 km pixels. After co-location, relevant MODIS cloud height values are gathered and averaged for each MOPITT pixel.

L258-L270: '4.4 MISR and MODIS height comparison'

This is a small section which describes for two parts of a scene the comparison between MISR and MODIS cloud height and concludes: 'Therefore, MISR and MODIS agree with the cloud height values.' I don't believe that such a conclusion can be drawn from the material presented here. Also, the related Figures 4 and 5 do not at all allow the reader to easily judge on this conclusion. I strongly recommend to skip this section and better provide and discuss references on previous MODIS cloud height validation.

Thanks for the comment, this section is removed in the revised manuscript.

Chapter 4.3, 4.5:

The related Figures (3, 6-9) as well as the discussion do provide only qualitative views and a first impressions of the situation. However, a quantitative analysis is entirely missing. E.g. one could discuss the different regions via scatter plots, MOPITT CO (with/without cloud-mask/only low clouds) versus MODIS cloud height/cloud cover and versus IASI CO.

Thanks for the comment, further quantitative analysis is added in the revised manuscript as illustrated in section 4.5 and 4.6. Section 4.5 presents a detailed correlation between the MOPITT radiance ratio (MRT) and MODIS cloud height over Canada and examines how adding the MRT cloud test independently in the V9 cloud detection scheme resolved the problem of low cloud miss-detection over land. This coverage enhancement results in a significant data coverage increase, especially over Canada. Section 4.6 presents the MOPITT V9, V8 and IASI CO TC analysis.

---

## Author Comment (AC2)

We would like to thank the reviewer for his/her insightful comments on the manuscript. Below is our response (black) to the referee comments (red).

RC2: 'Comment on amt-2021-112', Anonymous Referee #2, 26 Jun 2021

The authors present an overview of the MOPITT standard data product (clear-sky observations or over low clouds for ocean scenes). The authors present a quantitative study of including low cloud areas in the retrievals. The authors quantitatively show that MOPITT data were improved when low cloud areas were included in the retrievals. Given that MOPITT has measured CO since 2000 updating the current L2 data product is essential. Therefore, the authors need to address improving the current MOPITT data product to include low cloud areas in the retrievals. A section about validation and comparison between MOPITT, IASI, TROPOMI, and ground-based measurements is needed.

Based on the major issues highlighted below, I can't recommend the manuscript for publication in its current form. However, the authors can resubmit the manuscript if they address the major issues.

As we noted in our response to Reviewer 1 (and in our response below), there was some confusion as the focus of the manuscript. We have significantly revised the manuscript and have changed the title to better describe the unique contribution of this analysis to the evaluation of the MOPITT V9 product. In addition, this study shows comparisons with the IASI instrument that are not presented elsewhere.

**Major Issues:**

1. A quantitative study without any detailed analysis. Plots of the daily mean of total CO columns with and without low cloud areas included.

We have significantly revised the manuscript and we reduced the number of cases. Now the analysis of the impact of clouds on the observational coverage is much more quantitative analysis as new sections 4.5 and 4.6 are added.

2. Description of error sources and analysis.

The focus of the paper is not on MOPITT retrieval algorithm, so it would be beyond the scope of the work to go into a detailed discussion of retrieval error analysis that is discussed by Deeter et al. (2021).

3. A clear plan to adopt and improve on the current MOPITT L2 data product is missing. The authors did not state or propose to modify the current version of MOPITT data.

We apologize for the confusion; our plan was not to propose a new product. However, the paper aimed to investigate the issue of increasing the number of MOPITT observations and suggest a way of increasing the coverage rate in cloudy conditions. Work was being done in parallel with our colleagues at NCAR to produce a new product, version 9 (V9), and a description of this new product was published recently in Remote Sensing of Environment by Deeter et al., 2021. This new MOPITT V9 version product is available now to the public. As a result, we have restructured the manuscript to better reflect this and to complement Deeter et al. (2021). We have revised the paper to complement Deeter's work by providing a detailed analysis of the improvements in V9 MOPITT observational coverage. The analysis is conducted to understand the impact of cloud conditions on the MOPITT observational coverage of V9 and V8, with a particular focus on observations over Canada using Moderate Resolution Imaging Spectroradiometer (MODIS) cloud heights and cloud mask products along with MOPITT retrieval cloud flag descriptors.

4. Addition of validation and comparison between MOPITT, IASI, TROPOMI, and ground-based measurements section.

A quantitative comparison of MOPITT V9 TIR with the corresponding IASI CO is conducted for three cases. The first and third cases are associated with biomass burning emissions, while the second case represents typical conditions with no extreme air pollution. The study revealed positive bias of IASI relative to MOPITT which mostly occur at high CO values, and since the added data in V9 are mainly in heavily polluted regions, the IASI bias is greater for V9 than V8. Understanding the factors that potentially contribute to the discrepancies between MOPITT and IASI will be further investigated in future work.

the performance of the revised cloud detection algorithm is evaluated through validation based on a set of in-situ CO profiles acquired during the ACRIDICON-CHUVA campaign and NOAA Aircraft Profiles by Deeter et al., 2021.

Given the emphasis of the analysis on the impact of clouds on observational coverage, expanding the analysis to include TROPOMI would require a more challenging validation focus of the manuscript because of the differences in timing and spatial resolution of the MOPITT and TROPOMI measurements (morning, 10.30 am, compared to afternoon, 1.30 pm, and 22x22 km compared to 4x5km). The average kernels are also significantly different.

**Minor Issues:**

1. Low-quality images.

   We apologize for the confusion; the figures are repeated in the revised manuscript with better quality.

2. Some discussion of the physics of the retrievals can be beneficial.

As noted above, the focus of the paper is not on the MOPITT retrieval, so it would be inappropriate to go into a detailed discussion of the physics of the retrievals. Additionally, in the lines 50:60 of the manuscript the retrieval details are referred to in references such as Drummond et al. (1996), Drummond et al. (2010), and Deeter et al. (2003). However, the details of the MOPITT cloud detection algorithm are discussed in detail in section 3.

Lack of references (ex. L51).

References are added in line 49 in the revised manuscript.

Specific Comments:

1. Clarify L99.

It is rephrased for clarification. The statement "Consequently, adjustments to the current MOPITT cloud detection scheme is the only one of the four approaches that can be employed" in line 99 is now changed to "Consequently, adjustments or modifications to the current MOPITT cloud detection scheme is the only way to improve CO retrievals in cloudy conditions" in line 95.

2. L114 implies other products are available. If this is true, please elaborate.

This line (114) is removed in the revised manuscript.

---

## Author Response (AR2)

We would like to thank the reviewer for his/her positive and insightful comments on the manuscript. Below is our response (black) to the referee comments (red).

L41: 'a 5-10% positive bias that increases in highly polluted scenes.'

Here it should clearly be mentioned to which values the bias increases (up to >100% for highly polluted scenes). I don't think that mentioning such high numbers decreases the value of the dataset since it is even not clear if IASI or MOPITT or both are erroneous, if at all.

We meant that this bias is more pronounced in polluted conditions. The word "increases" is replaced by "more pronounced" as indicated in line 41.

L109:

On the request: 'Please specify in the instrument description the swath width, pixel size etc. which is referred to later on.' This answer has been provided: 'Thanks for the comment, it is added in lines 110:113 in the revised manuscript'
However, I can't see this in the revised manuscript.

We do apologize for this mistake. It was accidentally missed in the final version that we submitted.

It is added in the revised manuscript in lines 110-112.

L159: 'If' -> 'if'

Thanks for the comment, it is changed as it is shown in line 163

L210-L222: There is quite a confusion in this paragraph with figure numbers and 'left', 'right'…

We do apologize for the confusion. The figure numbers are clarified in the revised manuscript in lines                                                                                            215-229.

L259-L264:

Original comment: 'How is the MODIS cloud height for a MOPITT pixel determined (should also be described in the instrument-section)?'

Answer: 'Thanks for the comment, it is added in lines 259:264 in the revised manuscript.'

However, I cannot see any change here to the original manuscript. We apologize for the mistake. It is added in the revised manuscript in lines 269-274.

L330: 'It' -> 'it'

Thanks for the comment, it is changed as it is shown in line 349.

L343: 'the Canada region' -> 'the region of Canada'

Thanks for the comment, it is changed as it is shown in line 363.

L351: '7b' -> '10b'

Thanks for the comment, it is changed as it is shown in line 371.

L352: '7d' -> '10d'

Thanks for the comment, it is changed as it is shown in line 372

L353: can you comment on the smaller density of points in V9 compared to V8 which seems not to be the case in Fig. 12 and Fig. 14.

Thanks for the comment, the low density of V9 compared to V8 in Figure 10 is because those are the data that are collocated with IASI, which means the non-collocated V9 points are not plotted. So, to avoid confusion,

we have replotted the V9 MOPITT Figure 10 to include all data points and hence it does not have low density compared to V8.

L440: 'interesting'

How is 'interesting' defined? Please use some more descriptive expression.

Thanks for the comment, we meant by interesting, coherent structures of CO plumes. The word interesting is removed and replaced by coherent structures of CO plumes as indicated in line 463.

L457: There is something wrong with this sentence.
Thanks for the comment, the statement is rephrased to be more clear as indicated in lines 480-483.

L460: 'which increases in highly polluted scenes'
Same comment as for the same expression in the abstract: please be more quantitative here.

Thanks for the comment, the phrase " highly polluted scenes' is removed and the whole statement is rephrased to be more clear as indicated in lines 480-483.

Fig. 3, middle panel:

Could you explain why in this difference plot there are clear orbit-like structures extending from about 30 deg S to 30 deg N latitude between about

20 deg west and 140 deg east?

Thanks for the comment, the orbit-like structures is resulted from missing some days in April 2014. We examined all MOPITT years till 2020, and we found that 2014 is the only year that has an odd structure. We do apologize for accidentally choosing 2014.  To avoid this problem, We replotted Figures 2-5 using 2013 instead of 2014.

L294: 'left', 'right' -> 'top', 'bottom'

Thanks for the comment, it is changed as it is shown in line 623.

Figure 10 a,b,c: 'May.' -> 'May'

Thanks for the comment, it is changed as it is shown in Figure 10.

L634: 'Figure 7' -> 'Figure 10'

Thanks for the comment, it is changed as it is shown in line 662.

L641: 'Figure 3' -> 'Figure 10'

Thanks for the comment, it is changed as it is shown in line 669.